# Utility of a freehand frameless navigation system in computed tomography-assisted ventral bulla osteotomy for otitis media in calves

Takeshi Tsuka[1]*, Masamichi Yamahita[1], Yoshiharu Okamoto[2], Shunsuke Miyazaki[3], Jun Ishii[4], Kitaro Yoshimitsu[5], Yoshihiro Muragaki[6,7]

1 Joint Department of Veterinary Clinical Medicine, Faculty of Agriculture, Tottori University, Tottori, Japan, 2 WOLVES HAND Advanced Veterinary Medical Research Institute, Osaka, Japan, 3 Hyogo Prefectural Federation Agricultural Mutual Aid Association, Kobe City, Hyogo, Japan, 4 Himeji Livestock Hygiene Service Center of Hyogo Prefecture, Himeji-City, Hyogo, Japan, 5 Faculty of Advanced Techno-Surgery, Advanced Biomedical Engineering and Science, Tokyo Women's Medical University, Tokyo, Japan, 6 Department of Neurosurgery, Kobe University Graduate School of Medicine, Kobe, Japan, 7 Center for Advanced Medical Engineering Research & Development, Kobe University Graduate School of Medicine, Kobe, Japan

* tsuka@tottori-u.ac.jp

## Abstract

Ventral bulla osteotomy has not been widely adopted for treating otitis media in calves due to its prolonged surgical duration (>1 h), high surgical invasiveness, and challenges in application for bilateral cases. This study aimed to evaluate the utility of a freehand frameless navigation system for computed tomography-assisted ventral bulla osteotomy in 21 calves diagnosed with otitis media. Preparation times—including computed tomography examination and registration procedures—varied between 18 and 73 min. This technique allows for a minimal skin incision, approximately 1 cm in length, allowing instrument access via a 7-mm-diameter trocar. Surgical times for the 36 affected ears across 21 animals ranged between 7 and 26 min per ear. This technique reduced total procedure time, including preparation and surgical times. The median durations were 49.0 min (range, 41–57 min) for unilateral cases and 66.0 min (range, 47–106 min) for bilateral cases. In 13 of the 21 treated animals, otitis media-associated clinical signs improved or disappeared suddenly or gradually after surgery. Postoperative complications included generalized convulsive seizures and swollen mandibles in one and three animals, respectively. Areas for improvement warranting further development include: (1) prevention of unexpected complications and reduction of time-consuming preparation steps contributing to prolonged preparation time, (2) method for securing the calf's head on a surgical table, and (3) technical transfer from manual operation to utilization of an electric drill device to create a perforation om the affected tympanic bulla.

**Data availability statement:** All relevant data are within the manuscript and its Supporting Information files.

**Funding:** The author(s) received no specific funding for this work.

## Introduction

Otitis media is one of the most common infectious diseases in calves, typically occurring in pre-weaned individuals, and occasionally in post-weaned calves up to 18 months of age [1–3]. *Mycoplasma bovis* (*M. bovis*) is a common pathogen isolated in most bovine cases and is frequently found in co-infection with various bacteria, such as *Pasteurella multocida* (*P. multocida*), *Haemophilus somnus*, *Mannheimia haemolytica*, and *Escherichia coli* (*E. coli*) [1,3–10]. Other *Mycoplasma* species, such as *Mycoplasma bovirhinis* (*M. bovirhinis*) and *Mycoplasma dispar* (*M. dispar*) have also been isolated from bovine otitis media [1,6,7,11]. The Eustachian tube is considered a common entry point of infection, in which these causative pathogens can spread from the nasopharynx to the middle ear, whereas other routes may include hematogenous transfer systemically infected pathogens and migration from the external auditory canal [1,3,7,8]. The transfer of the pathogen from the nasopharynx to the middle ear can lead to concurrent otitis media and pneumonia [6,8]. Additionally, the infection route may cause spread of the pathogen toward both middle ears, resulting in bilateral otitis media/interna in calves [12].

Continuous and long-term administration of antimicrobials and anti-inflammatory drugs is the first choice of treatment for bovine otitis media, allowing better therapeutic efficacy in acute-phase lesions [13]. In Japan, macrolides and tetracyclines are the first-choice antimicrobials for mycoplasma infections [14,15]. However, the therapeutic effects of various traditionally used antimicrobials for mycoplasma infections have recently decreased [9,16–18]. Additionally, the administration of antimicrobials does not always resolve the clinical signs of chronic or complicated otitis media [2,8,13,19,20].

Ventral bulla osteotomy (VBO) is one of the surgical options for treating otitis media [12,13,19]. Briefly, an artificial perforation was made in the ventral walls of the affected tympanic bulla using a trocar, introduced through a surgical opening made in the ventral surface of the cranial neck, with the animal positioned in dorsal recumbency [12]. Deep anesthesia was required during this procedure. Thus, a shorter time required for surgery is recommended to reduce anesthesia-associated stress in treated animals, many of which suffer from varying degrees of respiratory disturbance accompanying otitis media [6,8]. However, the traditional VBO procedure is limited by its long duration, typically exceeding one hour [12]. Most of this time is spent developing the surgical approach, which is particularly challenging due to the need for sufficient macroscopic exposure of the tympanic bulla for perforation. This process involves wide retraction of the omohyoid and sternothyroid muscles using Gelpi retractors, which are difficult to perform quickly and carries a significant risk [12]. Compression caused by these retractors can induce iatrogenic vagus nerve injury leading to cardiac arrhythmia and sudden cardiac arrest [12]. Thus, enhancing the safety and efficiency of VBO in bovine cases necessitates technical innovations such as the implementation of image-assisted methods [12].

Imaging-assisted biopsy and surgical techniques have already advanced in human medicine through basic research using cadaver specimens [21], clinical trials for

various patients [22–25], or a combination of both [26, 27]. The imaging modalities used when performing these techniques include computed tomography (CT) [26–30] or magnetic resonance imaging (MRI) [23,25], or a combination of both [24]. The development of these techniques was strongly associated with the technical and mechanical evolutions of imaging-assistant systems that are mainly categorized into the following two devices: one includes various types of frame-based stereotactic (non-navigation) devices such as patient-customized frames mounted and anchored to the skull during brain biopsy [22], and the other is a navigation system using an infrared camera, categorized into frame-based or frameless systems [23,25,30,31]. The use of a frame-based stereotactic system can lead to a better application accuracy than a frameless navigation system [32,33]. However, frameless systems offer greater operational flexibility and shorter setup times, which has led to their widespread adoption in human medicine [28,30]. These imaging-assistant systems are currently utilized for otologic surgeries such as cochlear implantation, posterior tympanotomy, and the removal of various tumors such as vestibular schwannomas [21,22,24,26,27,29,34]. Since the 1960s, animal models have been used for the early development of stereotactic devices, including pigs and small ruminants, such as sheep and goats [35]. In veterinary medicine, two types of imaging assistant devices have been used since the 1990s [36]. Frame-based stereotactic devices have been predominantly used in previous experimental animal studies using cadaver specimens from dogs [32,33,36–42], pigs [43,44], goats [35,45], and sheep [46]. In parallel with experimental research, the clinical use of these devices has advanced to include brain biopsies [39,47–49], intracranial catheter placement [50], and radiosurgery [51–53] in anesthetized dogs and cats. According to current trends in human medicine [54], frameless navigation systems, referred to commonly as neuronavigation, have begun to be applied in experimental research involving dogs [54–61], cats [60,62], horses [63] and sheep [64,65], followed by the clinical utilizations for brain and nasal biopsy [66], the guidance of endoscopy-assisted surgical removal of a cholesteatoma [59], and ventricular shunt placement in the canines with hydrocephalus [67–69]. However, to the best of our knowledge, there are no previous reports describing the clinical use of frameless navigation systems for large animals, except for one experimental study using equine cadaver specimens [63]. Additionally, this is the first report to describe the utility of a frameless navigation system for directly guiding the treatment trocar into the affected tympanic bulla during VBO, in contrast to a previous canine report describing the indirect utilization of endoscopy, followed by its use of guidance for VBO [59].

The aims of the present report were (1) to methodologically explain a CT-assisted VBO technique using a frameless navigation system for bovine cases of otitis media, (2) to compare preoperative and postoperative medication protocols, and (3) to evaluate follow-up results, including postoperative complications, outcomes, and prognoses. The technical advantages and disadvantages of our method have been reviewed in previous human and veterinary reports.

## Materials and methods

### Animals

This study included 21 calves diagnosed with otitis media refractory to medical treatment and admitted to Tottori University. Of these, 20 were Japanese-Black (JB) beef calves and one was a JB-Holstein mixed-breed calf; 13 were males and 8 were females. Written informed consent was obtained from the owners prior to treatment. The present study protocol adhered to the ethical guidelines established by the Ethics Committee of the Faculty of Veterinary Medicine, Tottori University. Ethical approval was granted (https://assets.publishing.service.gov.uk/government/uploads/system/uploads/attachment_data/file/388535/CoPanimalsWeb.pdf).

### CT examinations

CT examinations were performed using a 16-section multidetector scanner (ECLOS; Hitachi Co., Ltd., Tokyo, Japan). Scans were acquired with a slice thickness of 0.625-mm and X-ray tube settings of 120 kVp and 175 mA. An image analysis system (AZE Virtual Place, AZE Corp., Tokyo, Japan) was used for the measurement of the tympanic bulla, including

dorsoventral height and mediolateral width on the transverse section, craniocaudal length on the dorsal section, and a maximum thickness of the ventral wall on the sagittal section of the skull CT [12]. Additionally, the maximum distance between the skin surface and the ventral wall of the tympanic bulla was also measured on the sagittal section. Preoperative CT was used for diagnosis and surgical planning. Intraoperative CT was used to confirm successful trocar placement in cases where purulent drainage was not macroscopically evident.

## Navigation system device

The navigation system employed in this study was a safety-supported model used for human brain tumor patients at Tokyo Women's Medical University, comprising a personal computer with neuronavigational software, a Polaris Vicra optical position sensor system (Northern Digital Inc., Waterloo, Canada), and a computer screen (Fig 1a) [70]. The optical sensor was mounted on an articulated arm, such that it could be directed from a high position toward the surgical table. A V-shaped plastic surgical table was used for CT scans and surgery. Four fiducial markers (Northern Digital Inc., Waterloo, Canada) were attached to the surgical table, with two markers located on each inner surface of the V-shaped table. A navigation pointer with three reflective spheres was used for reconstruction (Fig 1b). A securing device with three reflective spheres was fixed to the uppermost part of the sleeve of a trocar (outer diameter of the sleeve: 7 mm; length of the sharp stylet: 90 mm; Fujihira Industry Co., Ltd., Tokyo, Japan) (Fig 1c).

## Bacterial examination

Bacterial isolates were cultured from irrigation fluids using blood agar, chocolate agar, and deoxycholate-hydrogen sulfide-lactose agar. *Mycoplasma* species were identified by polymerase chain reaction (PCR) of isolates grown on *Mycoplasma* NK agar (Kanto Chemical Co. Inc., Tokyo, Japan).

## Clinical data

Pre-and postoperative therapeutic records were obtained from veterinarians of the Hyogo Prefectural Federation Agricultural Mutual Aid Association. To evaluate the surgical records, the times required for preoperative preparation and

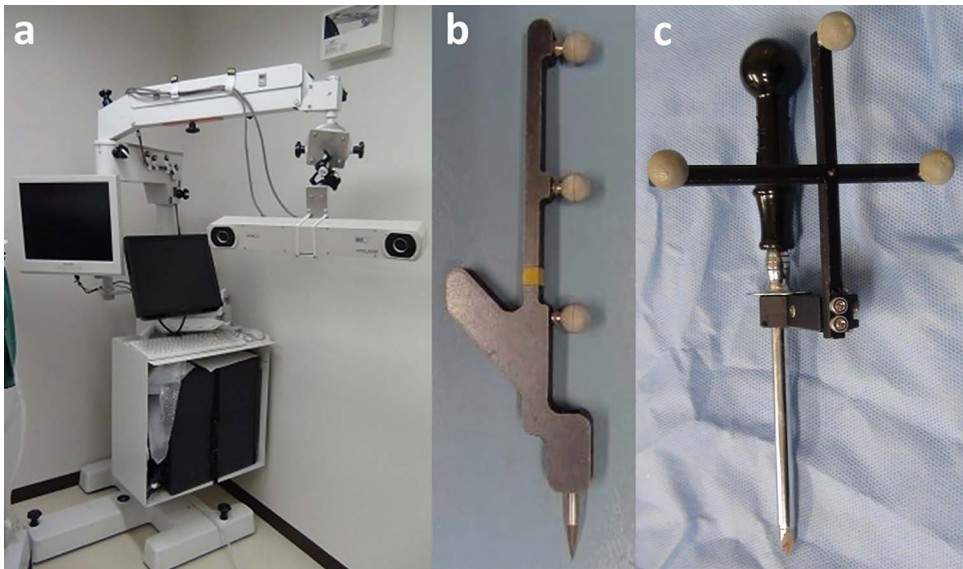

**Fig 1. Equipment used for the freehand frameless navigation system during computed tomography-assisted ventral bulla osteotomy.**

surgical procedures were used. Preparation time was defined as the time taken for preoperative CT scanning and transfer of the CT files from a CT machine to a navigation computer via digital recording media, followed by a registration procedure using a navigation pointer operated by a surgeon under aseptic conditions when surgery could already be started. Surgical time included the total time required for skin incision, tympanic bulla perforation, irrigation via a trocar, and skin closure. In cases of bilateral otitis media, the surgical time was recorded for the first and second surgical procedures performed on the left and right tympanic bulla, respectively. The total procedure time was calculated as the sum of preparation and surgical times. The number of trocars introduced into the affected tympanic bulla during each navigational procedure was evaluated. These data were obtained from the clinical records of 20 animals, excluding Case #1. Unexpected intraoperative events and complications were also recorded.

### Data analysis

The five measurement parameters of preoperative CT were statistically compared between the affected and unaffected tympanic bulla using Welch's t-test. The 21 animals were divided into two groups based on positive or negative surgical outcomes. Preoperative and postoperative clinical data were statistically compared between the two groups using Welch's t-test. Analyses included the number of medical treatments before surgery, interval from initial clinical presentation to surgery, and number of postoperative medications. The total procedure time was compared between the two groups using Welch's t-test. Additionally, a comparison of the total procedure time was performed between animals with bilateral (n = 14) and unilateral (n = 6) otitis media using Welch's t-test. Statistical significance was set at P-value < 0.05. significant.

## Results

All 21 calves initially presented with respiratory distress between 5 and 108 days of age and subsequently developed clinical signs of otitis media (Table 1). All animals had a history of 2–48 administrations of antimicrobials, including enrofloxacin (Baytril injection, Bayer HealthCare LLC), florfenicol (Nuflor, MSD Animal Health), kanamycin (Kanamycin sulfate injection, Meiji Seika Pharma Co., Ltd.), streptomycin–penicillin complex (Mycillin, Meiji Seika Pharma Co., Ltd.), marbofloxacin (Marbocyl 10% injection, Vétoquinol SA), oxytetracycline (Oxytetracycline injectable solution, Zoetis Inc.), tylosin (Tylan injection, Elanco Japan), and tulathromycin (Draxxin injectable solution, Zoetis JP).

Additionally, 14 of the 21 animals were treated with high-pressure irrigation of the tympanic cavity using a continuous drench gun (1–8 times) [12]. However, these medical treatments failed to resolve the clinical signs, and the animals were subsequently treated with VBO at an average of 33.9 days (2–83 days) after initial medical treatment.

### Surgical procedure

Animals were anesthetized using 2–3% isoflurane (Isoflo, DS Pharma Animal Health Co. Ltd., Osaka, Japan) via an endotracheal tube, inserted after sedation with xylazine hydrochloride (0.2 mg/kg, IV; Selactar 2%, Bayer Yakuhin Ltd., Osaka, Japan). The animals were positioned in dorsal recumbency on the examination table of the CT machine. The ventral areas of the face and cranial neck of dorsal recumbent animals were shaved and the animal was secured to a V-shaped plastic surgical table using surgical tape so that it was located between the four fiducial markers on the surgical table (Fig 2a). Disinfection was performed before CT examination. The areas of the face and cranial neck were scanned so that the four markers could be included in the same CT sections (Fig 2b). Otitis media could be diagnosed based on CT findings, including the accumulation of hyperattenuating materials within the cavity of the tympanic bulla with various degrees of extension and thickening and/or osteolytic changes in the bony walls [12,71].

The four fiducial markers (filled arrows) attached to the surgical table were imaged in the same places on the corresponding CT section (empty arrows). The locations of a 1-cm-length skin incision made on the ventral surface of the head and neck region (two lines) correspond to the locations of both affected tympanic bulla (arrowheads) on the CT.

**Table 1. Clinical data in the affected calves (n = 21) before surgical treatments.**

| Case | Age in initial onset (day) | Number of medications | Type of the used antimicrobials[a] | Number of ear irrigation | Age in surgery (day) |
|---|---|---|---|---|---|
| #1 | 75 | 28 | EF, FF, KM, MC, OTC | 2 | 140 |
| #2 | 93 | 2 | FF | 0 | 95 |
| #3 | 78 | 8 | MC | 4 | 101 |
| #4 | 89 | 14 | FF, KM, MC | 5 | 129 |
| #5 | 72 | 12 | FF, KM, MC | 8 | 100 |
| #6 | 108 | 7 | MC | 2 | 134 |
| #7 | 40 | 19 | EF, FF, KM, MC | 7 | 64 |
| #8 | 33 | 23 | EF, KM, MC, OTC | 2 | 66 |
| #9 | 38 | 14 | EF, MRFX, OTC | 1 | 53 |
| #10 | 65 | 10 | EF, OTC | 0 | 75 |
| #11 | 73 | 12 | EF, OTC | 1 | 85 |
| #12 | 47 | 15 | EF, KM, OTC | 0 | 64 |
| #13 | 44 | 4 | OTC | 0 | 48 |
| #14 | 9 | 40 | EF, FF, KM, MC, MRFX, OTC, TM | 1 | 89 |
| #15 | 5 | 48 | EF, FF, MC, MRFX, OTC, TL, TM | 2 | 88 |
| #16 | 25 | 16 | EF, MC, OTC, TM | 0 | 96 |
| #17 | 38 | 26 | EF, FF, MRFX, OTC, TM | 0 | 64 |
| #18 | 64 | 42 | EF, FF, MC, MRFX, OTC, TM | 1 | 133 |
| #19 | 102 | 42 | EF, KM, MRFX, OTC, TL, TM | 0 | 144 |
| #20 | 60 | 24 | EF, FF, KM, OTC, TM | 1 | 85 |
| #21 | 76 | 16 | EF, FF, KM, OTC, TM | 2 | 92 |

[a]EF: enrofloxacin; FF: florfenicol; KM: kanamycin; MC: streptomycin-penicillin complex; MRFX: marbofloxacin; OTC: oxytetracycline; TL: tylosin; and TM: tulathromycin.

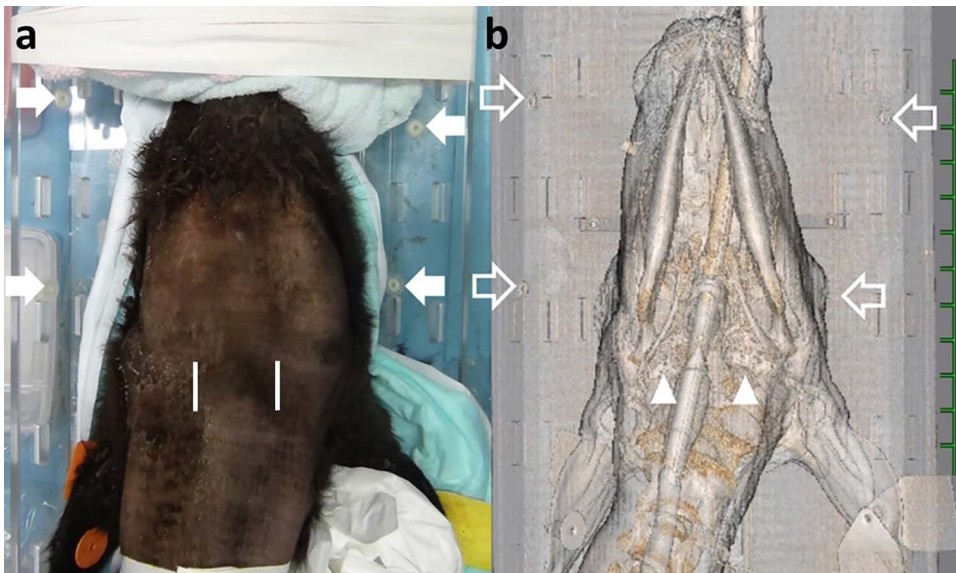

**Fig 2. Macroscopic view (a) and three-dimensional computed tomography (CT) image (b) of the head and neck region of a calf positioned in dorsal recumbency and secured on a V-shaped plastic surgical table.**

Preoperative CT identified bilateral otitis media in 15 animals and unilateral involvement in six (three in the left ear and three in the right ear) ([Table 2]). The dorsoventral heights, mediolateral widths, and craniocaudal lengths were measured at an average of 39.8 (32.2 to 51.3) mm, 34.2 (23.9 to 44.9) mm, and 39.1 (30.7 to 50.6) mm, respectively, in 36 affected tympanic bullae of the 21 animals. These three values were significantly ($p < 0.05$) higher than those in 6 unaffected tympanic bullae (average 34.8, 27.8, and 35.8 mm in dorsoventral heights, mediolateral widths, and craniocaudal lengths, respectively). Maximum thicknesses of the ventral walls in 24 affected tympanic bullae were measured at an average of 3.6 mm ranging 1.4 to 8.0 mm, as the values were significantly ($p < 0.05$) higher than those of the 6 unaffected tympanic bullae (an average of 1.5 mm, ranging from 0.8 to 2.6 mm). When comparing the distances between the skin and tympanic bulla, no significant difference was found between 36 affected and 6 unaffected tympanic bullae. The average distance for affected tympanic bullae was 54.5 mm (ranging from 43.8 to 73.9 mm), while for unaffected one it was 60.9 mm (ranging from 49.3 to 71.6 mm).

Soon after CT scanning, the original CT images were transferred into a navigation computer using a digital versatile disc. In a navigation computer, when the transferred original CT images were downloaded into a navigation software, the two-dimensional CT images, comprising the transverse, sagittal and dorsal views, were reconstructed. Subsequently, an integration procedure was performed as follows: (1) registration points were initially set at the center of the cross-sectional images of four fiducial markers, displayed on three reconstructed CT sections on the computer screen; (2) the left frontal fiducial marker was registered, while the sharp tip of the navigation pointer was placed in the center of the divot of the left frontal fiducial marker on the surgical table; and (3) this procedure was repeated for the left caudal, right caudal, and right frontal markers. The complete registration procedure was confirmed by the synchronous movement of the trajectory image on a computer screen corresponding to the tip of the navigation pointer when it moved near the location of the animal's face. Subsequently, a sterilized drape was placed on the animal's face and neck. The sharp tip of a trocar, secured in place by a device with three reflective spheres was subsequently registered by the surgeon; while the sharp tip of a navigation pointer was applied to the tip of the trocar and the shaft of the securing device within the sensitive area of an optical position sensor. Each location of the tip of the navigation pointer was registered into the navigation software.

VBO was initially performed for the left affected tympanic bulla, followed by surgical treatment of the right tympanic bulla for animals in which bilateral otitis media was diagnosed by preoperative CT. For all 21 animals, surgery was performed on the examination table of the CT machine in the CT room. The navigation system was placed on the opposite side of the dorsal recumbent animals by a surgeon standing on the left side of the treated animals ([Fig 3]).

The operator can handle a trocar fixed by a securing device with three reflective spheres (arrow) to create a perforation of the affected tympanic bulla while looking at the navigation computer screen.

The computer screen was placed in a position where the surgeon could easily look at it while performing VBO. Before skin incision, a trocar fixed by a securing device with three reflective spheres was used while moving slowly near the ventral skin surface at the location between the caudal face and the cranial neck to determine an adequate surgical approach point to introduce the trocar based on the trajectory image on a computer screen corresponding to its tip ([Fig 4a,b]). An approximately 1-cm skin incision was made at the planned location on the ventral skin surface. Using forceps inserted through the incision, blunt dissection was performed to create a surgical route for the trocar within the anatomical space between the skin surface and affected tympanic bulla. A trocar was introduced carefully through the skin incision while the surgeon operated the trocar so that the three reflective spheres on the attached securing device faced the optical position sensor. An operator advanced the trocar while looking at the computer screen, on which the trajectory image of the trocar tip passed along the inner side of the hyoid bone and was above the ventral wall of the tympanic bulla ([Fig. 4c,d]). After identifying the trajectory image on the surface of the tympanic bulla, the surgeon pushed the trocar forward on the affected tympanic bulla. Success in perforating the tympanic bulla was confirmed based on the location of the trajectory image of the trocar tip within the tympanic bulla cavity ([Fig 4e,f]). Perforation of the tympanic bulla walls was repeated if the trajectory image was not located within the cavity. Additionally, according to the surgeon's request for identification of the

**Table 2. Computed tomographic measurements of the tympanic bulla and the distance between the skin and tympanic bulla.**

| Case | Left/ right | Otitis media[a] | Dorsoventral height (mm) | Mediolateral width (mm) | Craniocaudal length (mm) | Maximum thickness of ventral wall (mm) | Distance between skin and tympanic bulla (mm) |
|---|---|---|---|---|---|---|---|
| #1 | Left | (+) | 50.9 | 38.5 | 47.0 | 6.0 | 49.3 |
| | Right | (+) | 48.6 | 46.7 | 49.2 | 4.3 | 55.7 |
| #2 | Left | (-) | 35.6 | 23.8 | 36.5 | 0.8 | 49.3 |
| | Right | (+) | 39.8 | 34.0 | 41.2 | 3.9 | 44.6 |
| #3 | Left | (+) | 34.7 | 28.8 | 33.9 | 1.8 | 61.5 |
| | Right | (+) | 44.8 | 37.3 | 34.5 | 2.4 | 60.1 |
| #4 | Left | (+) | 49.5 | 41.2 | 42.5 | 4.2 | 55.1 |
| | Right | (+) | 34.9 | 26.3 | 33.4 | 2.0 | 65.5 |
| #5 | Left | (+) | 39.8 | 30.8 | 39.7 | 3.2 | 56.0 |
| | Right | (+) | 37.3 | 35.2 | 44.1 | 3.9 | 55.1 |
| #6 | Left | (+) | 37.8 | 32.4 | 36.3 | 3.2 | 49.7 |
| | Right | (+) | 41.5 | 31.7 | 39.6 | 2.1 | 47.8 |
| #7 | Left | (+) | 40.9 | 36.1 | 45.4 | 4.2 | 63.1 |
| | Right | (+) | 40.7 | 38.1 | 40.6 | 5.1 | 64.0 |
| #8 | Left | (+) | 34.8 | 34.5 | 34.3 | 3.5 | 44.9 |
| | Right | (+) | 37.1 | 34.9 | 38.0 | 3.3 | 48.4 |
| #9 | Left | (+) | 36.0 | 29.6 | 34.9 | 3.8 | 52.4 |
| | Right | (+) | 35.6 | 28.7 | 32.8 | 2.7 | 58.9 |
| #10 | Left | (+) | 32.2 | 29.2 | 34.2 | 1.3 | 58.0 |
| | Right | (+) | 38.5 | 38.9 | 43.7 | 5.8 | 53.4 |
| #11 | Left | (-) | 35.4 | 24.4 | 35.1 | 1.1 | 61.1 |
| | Right | (+) | 43.6 | 31.5 | 40.4 | 7.5 | 58.2 |
| #12 | Left | (+) | 38.1 | 33.9 | 36.6 | 5.0 | 47.1 |
| | Right | (+) | 38.0 | 32.9 | 38.6 | 3.4 | 45.2 |
| #13 | Left | (+) | 32.4 | 23.9 | 30.7 | 2.2 | 49.6 |
| | Right | (+) | 30.9 | 26.3 | 30.8 | 1.6 | 56.6 |
| #14 | Left | (+) | 37.1 | 40.6 | 43.7 | 7.3 | 48.2 |
| | Right | (-) | 26.4 | 29.1 | 33.3 | 1.9 | 54.2 |
| #15 | Left | (+) | 40.2 | 35.6 | 41.0 | 7.4 | 45.8 |
| | Right | (+) | 33.6 | 32.8 | 37.6 | 2.2 | 60.8 |
| #16 | Left | (+) | 37.3 | 28.7 | 35.2 | 1.7 | 57.9 |
| | Right | (+) | 35.0 | 30.4 | 34.2 | 2.3 | 57.2 |
| #17 | Left | (+) | 39.6 | 32.9 | 36.6 | 4.0 | 43.8 |
| | Right | (+) | 36.3 | 28.7 | 34.3 | 1.9 | 50.4 |
| #18 | Left | (+) | 46.1 | 43.4 | 42.6 | 8.0 | 51.4 |
| | Right | (-) | 39.9 | 30.0 | 36.8 | 1.8 | 63.8 |
| #19 | Left | (+) | 51.3 | 44.9 | 50.6 | 4.5 | 58.0 |
| | Right | (-) | 34.2 | 26.3 | 35.5 | 2.6 | 71.6 |
| #20 | Left | (-) | 37.4 | 33.2 | 37.6 | 1.0 | 65.6 |
| | Right | (+) | 57.2 | 50.2 | 54.4 | 1.9 | 50.9 |
| #21 | Left | (+) | 41.1 | 32.0 | 41.1 | 2.1 | 64.4 |
| | Right | (+) | 38.6 | 28.2 | 33.8 | 1.4 | 73.9 |
| Affected tympanic bulla (n = 36) | | | 39.8±5.9 | 34.2±6.0 | 39.1±5.6 | 3.6±1.9 | 54.5±7.0 |
| Unaffected tympanic bulla (n = 6) | | | 34.8±4.6 | 27.8±3.6 | 35.8±1.5 | 1.5±0.7 | 60.9±8.1 |

[a](+) and (-) show the affected and unaffected tympanic bulla.

 

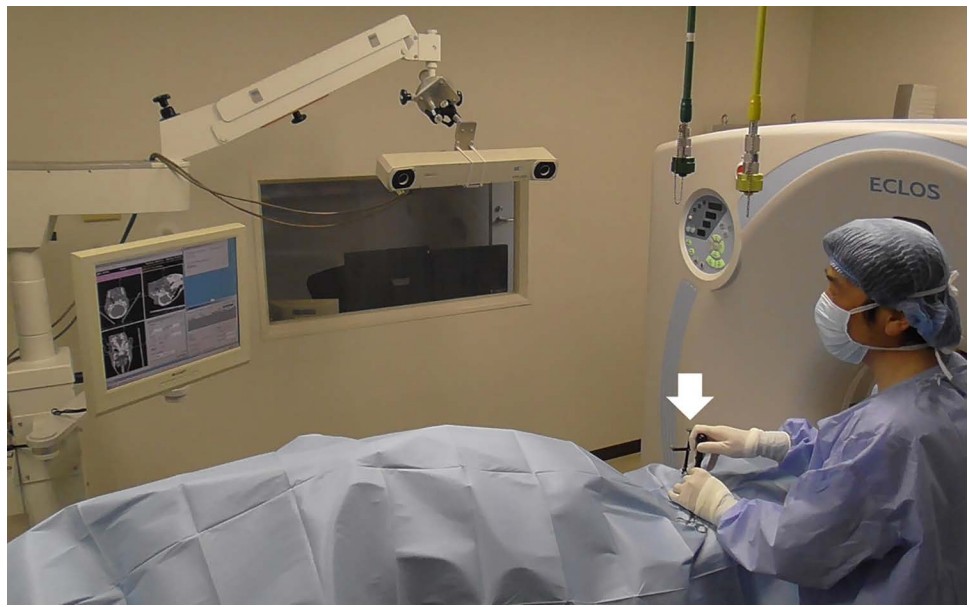

**Fig 3. Intraoperative photo of computed tomography-assisted ventral bulla osteotomy using a navigation system for a calf with otitis media.**

location of the trocar, the animal was examined using CT intraoperatively while holding the trocar in that position. Complete intra-ear introduction of the trocar could be identified on the intraoperative CT image, in which the line representing the tip of the trocar was present within the tympanic bulla cavity (Fig 4g,h).

A navigation computer screen can display the real-time location of the trocar tip, as represented by the lower part of the line on its screen. Intraoperative CT identified that a trocar fixed by a securing device with three reflective spheres was completely introduced within the tympanic bulla cavity. A dark-streaking metal artifact was observed on CT.

Through a plastic catheter (Atom pink catheter, 2.7 mm in outer diameter and 80 cm in tube length; Atom Medical Co., Tokyo, Japan) inserted via the trocar, 10 mL physiological saline solution was poured into the tympanic bulla. The irrigation fluid was then aspirated for bacteriological analysis. Based on the macroscopic identification of purulent materials floating in the collected fluids, a surgeon can confirm the complete procedure for placing a trocar into the tympanic bulla. The surgeon opted to either reposition the trocar or perform intraoperative CT scanning when no purulent material was identified in the irrigation fluids. Finally, via the catheter, the inside of the affected tympanic bulla was irrigated by alternating between flushing and collection using ozonated water provided by an ozonated water generator (Sakuragawa Pump Co., Ltd., Osaka, Japan) [72,73]. Irrigation was completed by identifying whether the irrigation fluids became clear without purulent deposits. After removal of the trocar and catheter, intradermic suturing was performed for the skin incision using an absorbable suture material (MAXON; Davis & Geck, USA). Drainage was not performed.

In the present VBO procedure, the time required for the preparation procedure ranged between 18 and 73 min (Table 3). The extended times in cases #9 and #13 were due to the need for two CT scans, caused by accidental displacement of the head secured with the surgical table. The amount of time required for surgery for the 36 affected ears of the 21 animals ranged between 7 and 26 min per ear. The extended duration in case #11 was caused by the need for two registration procedures, resulting from differences between the actual and imaged locations of the head during surgery. Numbers of trocar introduction per a procedure for 36 affected ears were either 1 or 2 times. The median total time, as calculated by the sum of the preparation and surgical times, was 49.0 min (range, 41–57 min) for the six animals with unilateral otitis media, and 66.0 min (range, 47–106 min) for the 14 animals with bilateral otitis media (except for case #1). There was a significant difference (p < 0.05) between the two values.

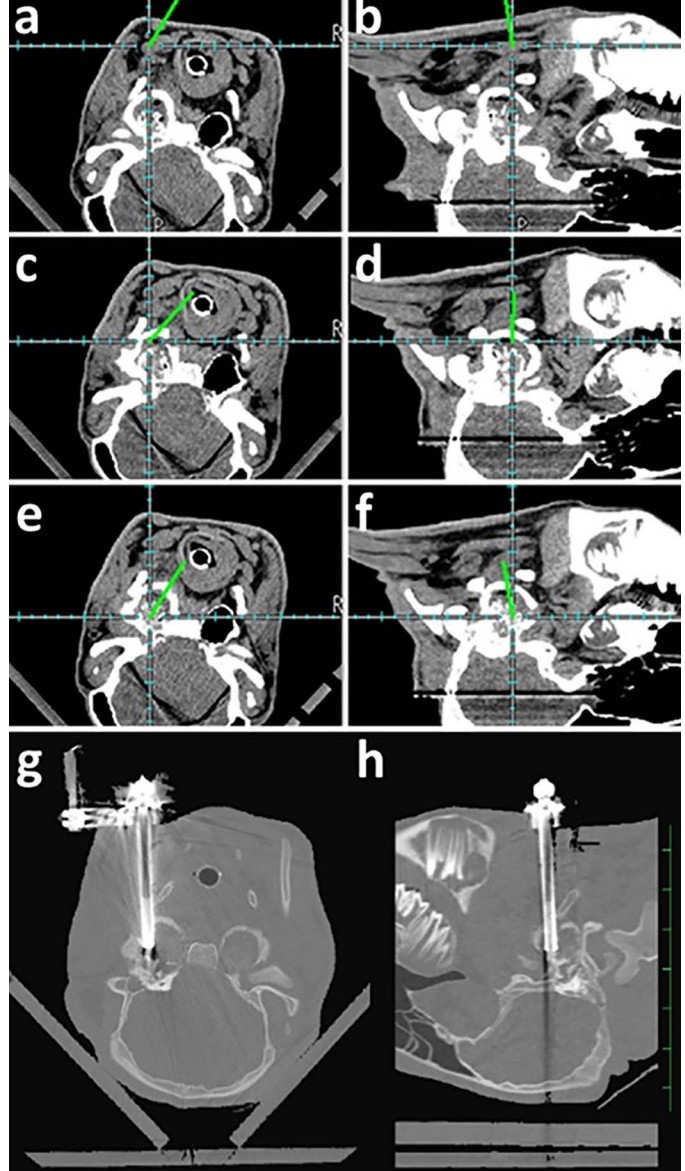

**Fig 4. Transversal (a, c, e) and sagittal (b, d, f) planes showing trajectory of the introduced trocar on a computer screen (case #18), and intra-operative computed tomography (CT) images (g, h) (case #7).**

*M.bovis* was isolated from all irrigation fluids obtained from 31 ears in 18 patients when examined bacteriologically (Table 4). *M. bovirhinis* was isolated from two of the 31 ears (right ears in cases #4 and #9). *M. dispar* was also detected in two of the 31 ears (right ear in case #4 and left ear in case #7). Three *Mycoplasma* species were identified in the same specimen from the right ear of patient #4. *E. coli* and *P. multocida* were isolated from four and two of the 31 ears, respectively, and co-infections with *M. bovis* were identified.

Of the 21 animals treated using the present VBO procedure, 20 received postoperative treatment consisting of 1–48 administrations of medication, including injections of up to 6 types of antimicrobials and 2–7 sessions of ear irrigation (Table 5). In case #20, a generalized convulsive seizure was observed soon after recovery from anesthesia, followed

**Table 3. Amount of time required and number of trocar introductions when performing ventral bulla osteotomy.**

| Case | Period of surgery | Affected side | Amount of time required (min) | | | | Number of trocar introduction per a procedure | |
|------|-------------------|---------------|-------------|---------------|---------------|---------|---------------|---------------|
| | | | Preparation | 1st procedure | 2nd procedure | Total | 1st procedure | 2nd procedure |
| #1 | Dec. 2015 | Left, Right | No record | No record | No record | No record | No record | No record |
| #2 | Dec. 2015 | Right | 43 | 14 | – | 57 | 2 | – |
| #3 | Dec. 2015 | Left, Right | 59 | 16 | 20 | 95 | 1 | 1 |
| #4 | Apr. 2016 | Left, Right | 45 | 11 | 9 | 65 | 1 | 1 |
| #5 | Apr. 2016 | Left, Right | 30 | 8 | 9 | 47 | 1 | 1 |
| #6 | Apr. 2016 | Left, Right | 66 | 10 | 10 | 86 | 1 | 1 |
| #7 | Jun. 2016 | Left, Right | 31 | 15 | 23 | 69 | 1 | 1 |
| #8 | Mar. 2017 | Left, Right | 52 | 9 | 14 | 75 | 1 | 1 |
| #9 | Mar. 2017 | Left, Right | 73[a] | 26 | 7 | 106 | 3 | 1 |
| #10 | Mar. 2017 | Left, Right | 38 | 10 | 16 | 64 | 1 | 1 |
| #11 | Mar. 2017 | Right | 18 | 26[b] | – | 44 | 1 | – |
| #12 | Apr. 2017 | Left, Right | 29 | 19 | 18 | 66 | 1 | 2 |
| #13 | Apr. 2017 | Left, Right | 48[a] | 19 | 14 | 81 | 2 | 1 |
| #14 | Apr. 2020 | Left | 31 | 10 | – | 41 | 1 | – |
| #15 | Apr. 2020 | Left, Right | 20 | 18 | 14 | 52 | 2 | 1 |
| #16 | Apr. 2020 | Left, Right | 42 | 13 | 11 | 66 | 1 | 1 |
| #17 | Apr. 2020 | Left, Right | 24 | 14 | 10 | 48 | 1 | 1 |
| #18 | Apr. 2020 | Left | 39 | 13 | – | 52 | 1 | – |
| #19 | Apr. 2020 | Left | 36 | 12 | – | 48 | 1 | – |
| #20 | Jun. 2020 | Right | 33 | 17 | – | 50 | 1 | – |
| #21 | Jun. 2020 | Left, Right | 40 | 9 | 7 | 56 | 1 | 1 |

[a]CT was scanned twice because of the accidental displacement of the head secured with the navigation table.

[b]Registration using a navigation pointer was performed twice because of the difference between the real and imaged location of the head.

by sudden death one day after surgery. In terms of therapeutic effects associated with surgical treatments, 13 animals showed sudden or gradual improvements in clinical signs associated with otitis media. However, two of these 13 animals (cases #4 and #13) died approximately 200 days after surgery due to progressive pneumonia. In the remaining seven animals, surgical therapy led to poor improvement of clinical signs associated with otitis media, and all seven animals died from pneumonia between 25 and 152 days postoperatively.

The clinical data were retrospectively compared between eight animals with unpreferable outcomes and 13 animals with preferable outcomes. The median number of medications administered before surgery was 23.5 (2–48) and 15.0 (4–76), respectively. Median intervals from onset of clinical signs to surgery were 29.5 days (2–83 days) and 24.0 days (4–80 days), respectively. The total time required for the VBO procedure was 54.5 min (48–106 min) and 64.5 min (41–95 min), respectively. The median number of postoperative medications administered was 25.0 (8–42 times) in seven animals with unpreferable outcomes and 14.0 (1–48 times) in 13 animals with preferable outcomes, except for case #20, which died on 1 postoperative day. No significant differences were observed between the two groups in any of the measured parameters.

## Discussion

In the present study, 21 animals did not show sufficient therapeutic response despite the continuous administration of multiple antimicrobials. Based on the present bacteriological results, intra-ear infections with *Mycoplasma* spp., mostly

**Table 4. Isolates in the irrigation fluids obtained from the affected tympanic bulla.**

| Case | Left | Right |
|------|------|-------|
| #1 | Not examined | Not examined |
| #2 | – | Not examined |
| #3 | Not examined | Not examined |
| #4 | *M. bovis, P. multocida* | *M. bovis, M. bovirhinis, M. dispar, P. multocida* |
| #5 | *M. bovis* | *M. bovis* |
| #6 | *M. bovis* | *M. bovis* |
| #7 | *M. bovis, M. dispar* | *M. bovis* |
| #8 | *M. bovis* | *M. bovis* |
| #9 | *M. bovis* | *M. bovis, M. bovirhinis* |
| #10 | *M. bovis* | *M. bovis* |
| #11 | – | *M. bovis* |
| #12 | *M. bovis* | *M. bovis* |
| #13 | *M. bovis* | *M. bovis* |
| #14 | *M. bovis, E. coli* | – |
| #15 | *M. bovis, E. coli* | *M. bovis, E. coli* |
| #16 | *M. bovis* | *M. bovis* |
| #17 | *M. bovis* | *M. bovis* |
| #18 | *M. bovis, E. coli* | – |
| #19 | *M. bovis* | – |
| #20 | – | *M. bovis* |
| #21 | *M. bovis* | *M. bovis* |

*M. bovis*, were the cause of otitis media in all animals. Reduced therapeutic efficacy against *Mycoplasma* infections is largely attributed to decreased antimicrobial susceptibility of the pathogens [9,16–18]. According to previously reported data on the antimicrobial susceptibilities of the drugs used in the present cases, streptomycin and penicillin complexes appear to have limited effectiveness, despite these drugs have been chosen. for the treatment of bovine otitis media [4,5]. However, the minimum inhibitory concentration of streptomycin required for killing *Mycoplasma* isolates is comparatively high, as β-lactam antibiotics, including penicillin, lack antimicrobial activity against *Mycoplasma* spp. [9,11,17]. *M. bovis* isolates in Europe were variably resistant to oxytetracycline, tylosin, and florfenicol, whereas a range of susceptibilities has been reported for fluoroquinolones such as enrofloxacin and marbofloxacin [9,17,18]. Despite the strong therapeutic effect of tulathromycin, its effects tend to vary among *M. bovis* isolates [18]. In Japan, a kanamycin-resistant phenotype has been detected in most *Mycoplasma* isolates, along with decreased antimicrobial resistance to oxytetracycline, tylosin, and florfenicol [14–16]. In terms of antimicrobial susceptibility to fluoroquinolones in Japanese isolates, there was great variation between the previous reports, as the prevalence of the resistant isolates was 12% [14–16]. On the other hand, unplanned and haphazard use of antimicrobials may have contributed to unfavorable therapeutic effects, compared with antimicrobial-resistant states. In 14 of the 21 animals, the affected ears were treated with at least one session of high-pressure irrigation using a continuous drench gun, an advanced intra-ear irrigation technique contributing to the varying healing rates ranging between 53.3 and 73.3% [12]. This technique aims to promote forced discharge of puru-lent materials from the tympanic bulla to the nasal cavity through the eustachian tube by applying high-pressure flushing through the auditory canal [12]. However, insufficient discharge could have occurred in the 14 animals treated with this technique. The prolonged use of unsuccessful conservative therapies could induce the development of chronic or end-stage otitis media that is difficult to treat [12,20]. Almost all of the present cases were considered to have chronic otitis media when being treated surgically, because the durations of medical treatments were extended to an average of 33.9

Table 5. Clinical data and prognosis in the affected calves (n = 21) after surgical treatments.

| Case | Number of medication | Type of the used antimicrobials[a] | Number of ear irrigation | Postoperative complications | Therapeutic effect | Age in death (postoperative days) | Cause of death |
|------|------|------|------|------|------|------|------|
| #1 | 5 | KM, MC, OTC | 0 | | Good | 983 (843) | Sold |
| #2 | 8 | EF, MC | 0 | | Poor | 120 (25) | Pneumonia |
| #3 | 1 | EF | 0 | | Good | 944 (843) | Sold |
| #4 | 6 | FF, KM, MC, OTC | 0 | | Good | 334 (205) | Pneumonia |
| #5 | 4 | FF, KM | 2 | | Good | 913 (813) | Sold |
| #6 | 14 | EF, FF, KM, MC | 0 | Mandibular swelling | Poor | 201 (67) | Pneumonia |
| #7 | 19 | FF, KM, MC, MRFX | 3 | Mandibular swelling | Good | 1112 (1048) | Sold |
| #8 | 26 | EF, FF, MRFX, OTC | 0 | | Poor | 93 (27) | Pneumonia |
| #9 | 25 | EF, FF, MRFX, OTC | 0 | | Poor | 78 (25) | Pneumonia |
| #10 | 14 | EF, FF, KM, OTC | 0 | | Good | 1027 (952) | Sold |
| #11 | 4 | EF | 0 | | Good | 1067 (982) | Sold |
| #12 | 7 | EF, OTC | 0 | | Good | 981 (917) | Sold |
| #13 | 38 | EF, MRFX, OTC | 2 | | Good | 240 (192) | Pneumonia |
| #14 | 20 | EF, KM, MC, MRFX, OTC | 0 | | Good | 955 (866) | Sold |
| #15 | 42 | EF, FF, KM, MC, MRFX, OTC | 0 | Mandibular swelling | Poor | 240 (152) | Pneumonia |
| #16 | 24 | EF, FF, KM, MC, MRFX, OTC | 0 | | Good | 1005 (909) | Sold |
| #17 | 38 | EF, KM, MC, MRFX, OTC | 0 | | Poor | 102 (38) | Pneumonia |
| #18 | 33 | EF, FF, KM, MRFX, OTC | 2 | | Poor | 243 (110) | Pneumonia |
| #19 | 12 | KM, MC, OTC | 0 | | Good | 931 (787) | Sold |
| #20 | 0[b] | | 0[b] | Seizure | Poor | 86 (1) | Seizure[c] |
| #21 | 48 | EF, FF, KM, MC, MRFX, OTC | 7 | | Good | 998 (906) | Sold |

[a]EF: enrofloxacin; FF: florfenicol; KM: kanamycin; MC: streptomycin-penicillin complex; MRFX: marbofloxacin; and OTC: oxytetracycline.

[b]Non-treatment, because of sudden death at one day after surgery.

[c]Exhibition of generalized convulsive seizure soon after recovery from anesthesia, followed by sudden death at one day after surgery.

days. Additionally, in preoperative CT, thickening and extension in the tympanic bulla's walls were the typical CT characteristics indicating the chronic phase [12,20]. Thus, VBO was chosen as an alternative therapeutic option for the present cases because 15 of the 21 patients had bilateral otitis media. This technique is suitable for treating both ears without the need for repositioning, whereas in lateral bulla osteotomy, the recumbent position must be changed for each procedure such that the affected ear faces upward [12]. However, it is necessary to modify the VBO innovatively to shorten the surgical time dramatically when applied to both ears.

The concept of the CT-assisted VBO technique may have been inspired by the results obtained from methods in otologic surgery in human medicine, in which the techniques could contribute to preoperative planning or real-time guidance to decide the trajectory of drilling, allowing the creation of a surgical tunnel toward the cochlea during cochlear implantation [21]. This drilling procedure was considered applicable to trocar operations using our method. Similarly, the trocar operation corresponds to the method of introducing a biopsy needle when performing an imaging-assisted procedure of brain biopsy and electrode placement, allowing vagal nerve stimulation within the brain [40,56]. In basic or clinical research on animal models and patients, the imaging modalities commonly chosen for frame-based stereotactic or frameless navigation systems are CT [36,40–42,46,48,50,51,53,54,59,68], MRI [32,43,44,47,55–57,62,65] alone and both CT and MRI [33,34,38,49,52,58,60,61,66,67,69]. CT is used more frequently than MRI for brain biopsies [55]. In imaging-assisted otologic surgery in human medicine, the clinical use of CT is predominant over the use of MRI [21,22,26,27,29].

This is because CT is more suitable for the creation of a skull opening and perforation because it allows clear visualization of bony structures and calcified lesions [24,70]. CT has been previously chosen as the advanced imaging modality for diagnosis and intraoperative evaluation of bovine otitis media [2,3,12,13,19]. Thus, CT is considered the most appropriate imaging modality for imaging-assisted VBO techniques in the bovine cases.

When advancing a biopsy needle and drilling equipment toward the targeted locations of the brain or ear, respectively, during imaging-assisted brain biopsy or otologic surgery, such equipment can be operated using various customized or commercial instruments to hold them [29] or freehand [21,25–27]. Holding instruments include stereotactic devices categorized into mounted platform types [33,36,40,48–50], articulated arm types [46,55], and bite plate types [42]. Some systems utilize various types of facemasks, within which a trajectory sleeve planned preoperatively was created [37–39]. These devices are categorized as frame-based stereotactic devices [37–39]. When using a navigation system, the use of holding instruments may lead to a more accurate trajectory planning performance than freehand operation [28,54]. However, the freehand technique is superior to the use of holding instruments in flexible operations [23,39,48,49,54,60,68], although it allows for a lower targeting accuracy for smaller and/or deeper lesions [28,61]. If holding instruments are used in the present VBO procedure, its fixed surgical route might not always be sufficient for trocar introduction because of the individual variety of skull sizes and shapes and the different degrees of extension of the affected tympanic bulla [12]. Thus, the freehand frameless navigation technique is considered acceptable for the proposed VBO technique.

Similar to developments in human medicine, advances in navigation systems in the veterinary field have enabled the application of freehand techniques in CT- or MRI-assisted procedures such as brain or nasal biopsy, intracranial catheter placement, and ventricular shunt placement [50,56,68,69]. The present freehand technique resembles a previous freehand technique used in CT-assisted biopsy, in which an already-registered biopsy rongeur inserted through one burr hole was initially created to collect multiple specimens, while its trajectory was monitored on the navigation screen [68]. The freehand technique has also been effectively used for intraoperative decision-making regarding the most optimal approach point for skull osteotomy, allowing complete surgical removal of skull base tumors while applying a pointer around the surgical areas [62]. The freehand method was applied to determine where the skin incision was to be made in the first step of the present VBO procedure.

When freehand frameless navigation techniques are applied to the human skull, the safety of the procedure cannot be fully guaranteed for lesions located more than 5 cm beneath the surface [25]. This is because the locational differences between the needle placement and targeted points, referred to as a target registration error or needle placement error, seem to be more dependent on the larger depths of the targeted structures, whereas some previous reports have described no correlation between errors and target depths [25,33,38,39,55]. Based on the CT measurements, the ventral surface of the tympanic bulla is mostly located more than 5 cm below the neck skin in dorsal recumbent calves treated with VBO. This was one possible cause why two or more trocar introductions were required in 5 of 34 ears in 20 cases except for case #1 (not recorded). Minimizing a target registration error within an accuracy of <0.5 mm is strongly required when performing imaging-assisted otologic surgery [34]. However, the freehand technique can generate larger errors between 1.0 and 4.0 mm, maximum 8.3 mm [34]. In previous studies using frame-based stereotactic and various navigation systems for brain biopsies, average needle placement errors ranged between 0.9 and 4.4 mm [32,33,36,39,41,42,44,49] and between 0.8 and 4.6 mm [54–57,60–62,65], respectively. These error ranges were considered acceptable for brain biopsies in veterinary medicine [60]. These errors seem to compound with multiple errors such as registration error [32,61] and spatial tracking errors of an infrared camera system, ranging between 0.2 and 0.5 mm [34,58]. The errors may be larger depending on the increase in slice thickness in the CT scan setting [34,40,42]. The slice thicknesses for CT scans in human and animals heads were 0.2 mm [34], 0.5 mm [27,39,58], 0.6 mm [26,54,68], 0.625 mm [46], 0.8 mm [22], 1 mm [21,49,52,53,61,63,66], 2–3 mm [41,48], and 4 mm [40]. The present VBO technique was performed using CT images with a slice thickness of 0.625 mm. A shorter CT slice thickness (0.2 mm) may be required

[34]. However, such settings would increase the number of CT images required to scan the entire bovine skull, resulting in a greater preoperative preparation time.

Accidental shifting of fiducial markers attached to the patient's body is another potential source of error [55,58,61]. The acceptable error was estimated at <1.2 mm when identifying the divots of the shifted fiducial markers using a navigation pointer [58]. Various stereotactic systems such as patient-customized face masks and bone-anchored devices can minimize accidental movement between the device and body because they facilitate rigid fixation, whereas earlier stereotactic devices used ear bars to secure the heads causing 2-to-3 mm displacements in relation to the coordinated location [33,35,39,45,47,60]. When using various navigation systems, the optimal method for placing fiducial markers should be considered for each technique because the use of bone-implanted fiducial markers can lead to better registration accuracy than the use of adhesive fiducial markers placed on the skin [26,55,57,62,74]. Bone-implanted fiducial markers have been applied to the heads of cadavers of experimental animals [63,65]. However, bone implantation of fiducial markers seems to have great disadvantages in terms of clinical utility, including surgical invasiveness and excessive time required in its setting [22,55]. A slightly invasive method for setting fiducial markers may include the placement of K-wire pin markers on the canine skull [61,66], and the use of a titanium screw and plastic cylinder filled with diluted gadolinium on the feline skull [62]. The bite-plate types of the fiducial marker's platform have been commonly used [54,55,58,68], This setup is unlikely to be applicable for the present VBO procedure in which the animals were positioned at dorsal recumbency. Thus, the use of a V-shaped surgical table with fiducial markers was chosen for the present VBO technique because it could effectively maintain the head position near the fixed fiducial markers. In the present VBO procedure, a trocar might be introduced into the targeted tympanic bulla with an error margin of several millimeters generated by various factors. including the setting of fiducial markers. However, the affected tympanic bulla, which is 3–4 cm in length and width, may have a target size where that level of error is acceptable.

The technical disadvantage of the present VBO technique includes metal-associated artifacts interfering with CT images when scanning during surgery because the trocar is made of iron. Metallic materials can generate bright and dark streaking artifacts distributed around them, masking the surrounding structures on CT [75]. In intraoperative CT scanning, to identify the successful introduction of the steel trocar within the affected tympanic bulla, metal artifacts could make frequent observation of its location difficult. Thus, the use of a trocar made from different materials, such as titanium, helps minimize metal artifacts, which can be reduced slightly by changes in slice thickness and the reconstruction method [26,74,75].

The present VBO procedure contributed to improvements in the clinical signs of otitis media in 13 of 21 treated animals. The therapeutic efficacy of VBOs in bovine cases is unknown because no retrospective reports have described the healing rate and morbidity of surgically treated animals. A key positive factor contributing to favorable therapeutic results, include reduced surgical time for each affected ear, which ranged from 7 to 26 min. These surgical times were notably shorter compared to the approximately 1 h required for traditional VBO, including skin incision, subsequent irrigation of the affected tympanic bulla, followed by skin closure [12]. For bovine cases, the traditional VBO procedure requires a > 10 cm skin incision because a wider surgical opening allows for deep observation of the affected tympanic bulla [12]. Larger skin incisions and surgical wounds extend the surgical time and enhance the severity of surgical invasiveness [24]. In the present VBO procedure, the required skin incision was < 1 cm in diameter, through which a trocar could be introduced. In our experience, the traditional VBO technique requires time-consuming work to create a route that allows macroscopic exposure of the affected tympanic bulla before its perforation [12]. The present procedure eliminates this, as the bulla can be perforated blindly. Additionally, based on the results in Table 3, the total time of the proposed VBO technique tends to decrease with experience. This is a common tendency observed when performing imaging-assisted brain biopsies and catheter placement procedures [50,54,60]. These factors can contribute to shortened surgical times.

In therapy for otitis media, water or saline is commonly recommended as the irrigation fluid to pour within the tympanic bulla, because it is a physically displacing material [76]. The addition of antimicrobial components and pH control to adjust

acidity in the irrigation fluids potentially leads to ototoxicity [76]. However, intraoperative irrigation of ozonated water into the affected tympanic bulla via a trocar might lead to supplementary therapeutic effects that act synergistically with the surgical effects. Ozonated water has the function of being inactivated, resulting in conversion to plain water soon upon contact with any organism, including microorganisms [72,73]. Thus, ozonated water is widely used as irrigation fluid in human medicine [72,73]. Ozone has potential microbicidal activity, including damaging the bacterial cell membrane and targeting cell wall components, peptidoglycans, cell surface antigens, and lipopolysaccharides [77,78]. This function can increase cell membrane permeability, resulting in cell lysis and leakage of cellular components [77,78]. Additionally, anti-inflammatory effects are another function associated with ozonated water, contributing to the local accumulation of superoxide dismutase induced by its oxidative activity [72]. However, the microbicidal activity of this material against *Mycoplasma* species is not fully known, whereas a previous laboratory test showed dramatic killing effects of ozone gas on *Mycoplasma bovigenitalium*, which is the causative agent of metritis in cows [78]. Ozone may have different functions in cell wall damage because of the lack of peptidoglycan in *Mycoplasma* [6]. The therapeutic potential of ozonated water for intra-ear irrigation warrants further clinical investigation, including applications during myringotomy and intraoperative procedures, and comparison with water or saline as a common material in intra-ear irrigation [76].

VBO is a surgical technique used to create a unilateral perforation of the tympanic bulla, regardless of traditional or CT-assisted procedures [12]. Thus, the surgical time is doubled for traditional VBO when treating both the left and right ear of animals with bilateral otitis media [12]. In contrast, the shortened surgical time per ear in the present CT-assisted VBO procedure contributed reduced overall surgery durations for both ears, ranging from 16 to 38 min in 15 of 21 animals. However, the total time taken for preparation and surgical procedures varied among individuals, with median values of 49 and 66 min for 6 and 15 animals with unilateral and bilateral otitis media, respectively. This depended on the time required for preparation.

Minimizing the time required for CT scanning and registration is difficult [25,68]. In a previous frameless navigation brain biopsy in human patients, setup times were >50 min, with a surgical time of 85 min [25]. The preparation time could be unexpectedly extended by the accidental shift of the head in two present cases. In another case, an extended surgical time required two registration procedures accidentally. Navigation procedures and transformation of animals between rooms are common causes of slight body movements, resulting in an increase in registration error [54,58]. Thus, in the present VBO procedure, bovine cases were treated following examination in a CT room without transfer, to prevent this accidental shift. However, disinfection and surgical procedures can lead to a slight rotation of the head and neck region because it is only secured on a V-shaped surgical table using surgical tape. Manual operation of a trocar to create perforation of the osseous walls of the tympanic bulla could generate intrafractional motion. The intrafractional motion can be induced when stronger pressure in the manual operation of a trocar is required, depending on the thickening of the tympanic bulla's walls, as measured at greater than 3 mm in Table 2. Thus, unexpected extension of the preparation and surgical time can be avoided by eliminating the possibility of intrafractional motion. The use of an electric drill device instead of a manual procedure may be effective in reducing intrafractional motion based on the preoperative estimation of tympanic bulla hardness using a Hounsfield unit on CT. Additionally, instead of the fiducial markers, the anatomical landmark may be applicable in the present VBO procedure [31,57,60,63,69].

Unfavorable outcomes in seven of 21 animals treated with the present VBO procedure were attributed to respiratory distress caused by progressive pneumonia rather than the poor therapeutic effects of VBO. In the seven animals, the relatively short interval between surgery and death (between 25 and 67 postoperative days) might have been caused directly by the compensatory exacerbating factors of both surgical invasiveness and respiratory load associated with deep anesthesia, facilitating increased damage to the lung structures in which infectious inflammation has already developed [6,8]. The present VBO technique is considered to reduce anesthesia-associated respiratory loads compared to the traditional VBO technique, although the total time required for anesthesia during surgery could not be obtained in the present report. Retrospective evaluation of the present clinical data suggests that delayed surgical intervention following ineffective

 

conservative therapies may have contributed to poor outcomes. In terms surgery time, in the seven animals with unpreferable outcomes, the median clinical course was 33 clinical days with a median administration of 23 times, compared to 24 days and 15 doses, respectively, in the 13 animals with preferable outcomes. This delay may have induced the development of the chronic phase of otitis media, causing refractory states, regardless of whether they were treated medically or surgically [8,12,20]. This problem also increased the number of postoperative medications needed in the unpreferable group, with a median of 25 doses compared to 14 in the preferable group. Chronic otitis media frequently requires prolonged medical treatment; several weeks of medication were previously required for clinical recovery [1,5]. Thus, similar to the seven unpreferable animals, the 13 preferable animals might have already had chronic otitis media. The present report could not provide definite evidence to influence the postoperative outcomes in affected calves treated with the present VBO technique, whereas the total time required for this surgery seemed to be less associated with the therapeutic effects.

Case #20 presented with a generalized convulsive seizure that resulted in sudden death. Osteolysis can commonly occur in the tympanic bulla affected by otitis media [3,12,20,71]. The osteolytic changes may also develop concurrently in the osseous labyrinth, which is the cavity of the temporal bone enveloping the inner ear structures. In the VBO procedure, this structure is located on the extension of the surgical route in which an operator introduces a trocar into the middle ear. In a manual operation to perforate the osseous walls of the tympanic bulla, excessive pressure on a trocar gripped by an operator could possibly lead to the destruction of the fragile, affected bone, causing accidental damage to the inner ear and the deeply located brain structure. Therefore, care should be taken when perforating the affected tympanic bulla. Mandibular swelling was a common postoperative complication, observed in 3 of the 21 animals. This might be caused by the subcutaneous retention of fluid that drained through the surgical perforation of the tympanic bulla, as well as the local inflammatory reaction of the soft tissue structures peripheral to the surgical wound. Holding a catheter in the treated tympanic bulla is recommended when performing VBO in bovine cases [12]. If intra-ear catheterization is performed using the present VBO technique, the catheter must be inserted into the tympanic bulla via the trocar while being introduced into the cavity during surgery. However, it is difficult to perform intra-ear catheterization blindly. Additionally, the drainage effect of the catheter held on perforations made downward within the treated tympanic bulla is debatable [12]. In the present report, two of the three calves did not show sufficient surgical effects. The postoperative prevalence of mandibular swelling and seizures should be further assessed based on the increased number of bovine cases treated using the present VBO technique.

The peak prevalence of *M. bovis*- associated otitis media has been previously reported to occur between 2 and 18 weeks of age [1,7,10,71]. Calves within this age range generally have moderate body sizes, allowing them to be maintained in dorsal recumbency during surgery. However, the onset of otitis media can sometimes be observed in calves aged >18 weeks of age [1,4]. The present VBO technique is not applicable to these animals because of the difficulty of maintaining them in dorsal recumbency. Instead, it is necessary to adapt the freehand frameless navigation system to CT-assisted lateral bulla osteotomy, a procedure that can be performed with the animal in lateral recumbency, so that any bovine case of otitis media can be treated surgically using this system.

## Conclusion

Imaging-assisted surgery using a frameless navigation system is still in the early stages of practical use, whereas biopsy techniques have already been developed as advanced methods in the veterinary field. The targeted structure in this technique is commonly the head region, based on the high clinical applicability of various otologic surgeries in human patients and brain biopsies in canine and feline cases. The structure of the head, within which the skull skeleton predominantly occupies thin soft tissue layers covering its surface, is suitable for firm fixation of the treated heads and stable setting of the navigation system in its region. In addition to this advantage, multiple factors contribute to the successful implementation of CT-assisted VBO using a navigation system in affected calves, including (1) the adequate size of the tympanic

bulla compensating for larger targeting errors, (2) the absence of obstructive anatomical structures along the trocar's trajectory when introduced from the ventral head-neck regions, (3) applicability of manual trocar operation for both insertion and perforation of the tympanic bulla, and (4) reduced surgical invasiveness, resulting in reduced surgical time. This knowledge can significantly contribute to further developments in imaging-assisted surgery in the veterinary field.

## Supporting information

**S1 Move. Surgical procedure of computed tomography-assisted ventral bulla osteotomy using a navigation system.** The operator handles a trocar fixed by a securing device with three reflective spheres such that these reflective spheres can face an optical position sensor during the surgical procedure to determine the skin incision, followed by the creation of a perforation of the affected tympanic bulla while looking at the navigation computer screen. It took 2 min 22 s to confirm completion of tympanic bulla perforation as confirmed by purulent materials aspirated into a plastic catheter inserted via a trocar.
(MP4)

## Acknowledgments

The authors would like to thank the veterinarians at Hyogo Prefectural Federation Agricultural Mutual Aid Association (Miyuki Baba, Takehisa Hirai, Shimesu Nishiguchi, Keisuke Morimoto, and Yasunori Suga) for their valuable clinical support.

## Author contributions

**Conceptualization:** Takeshi Tsuka.

**Data curation:** Takeshi Tsuka, Masamichi Yamahita, Yoshiharu Okamoto, Shunsuke Miyazaki, Jun Ishii.

**Investigation:** Takeshi Tsuka, Masamichi Yamahita, Yoshiharu Okamoto, Shunsuke Miyazaki, Jun Ishii.

**Methodology:** Takeshi Tsuka, Masamichi Yamahita, Yoshiharu Okamoto, Kitaro Yoshimitsu, Yoshihiro Muragaki.

**Software:** Kitaro Yoshimitsu, Yoshihiro Muragaki.

**Supervision:** Takeshi Tsuka.

**Writing – original draft:** Takeshi Tsuka.

**Writing – review & editing:** Takeshi Tsuka.

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
