## [Decision Letter · Decision Letter 0]

25 Nov 2025

Dear Dr. Tsuka,

Thank you for submitting your manuscript to PLOS ONE. After careful consideration, we feel that it has merit but does not fully meet PLOS ONE’s publication criteria as it currently stands. Therefore, we invite you to submit a revised version of the manuscript that addresses the points raised during the review process.

We look forward to receiving your revised manuscript.

Kind regards,

Toru Miwa

Academic Editor

PLOS ONE

**Journal Requirements:**

2. We note that Figure 3 includes an image of a participant in the study.

Reviewers' comments:

Reviewer's Responses to Questions

**Comments to the Author**

1. Is the manuscript technically sound, and do the data support the conclusions?

Reviewer #1: Yes

2. Has the statistical analysis been performed appropriately and rigorously?

Reviewer #1: Yes

3. Have the authors made all data underlying the findings in their manuscript fully available?

Reviewer #1: Yes

4. Is the manuscript presented in an intelligible fashion and written in standard English?

Reviewer #1: Yes

Reviewer #1: As discussed, the operational difficulty with the freehand frameless navigation system is influenced by the target size and its distance from the skin, which should be presented. From the preoperative CT scans, the following details should be included:

1. List the maximum diameter of the target bulla, including the horizontal and vertical section sizes, and the direct diameter if possible.

2. If available, describe the distance from the skin surface to the front wall of the bulla.

3. Note any diagnoses of middle ear disease, such as exacerbation of chronic otitis media or cholesteatoma otitis media infection, made through visual examination before or during surgery.

4. The presence or absence of sacral bone destruction and the extent of inflammation progression to surrounding tissues significantly impact the postoperative course. Therefore, these factors should be considered in the preoperative CT scan.

5. On page 43, line 641: It is important to note that while dizziness may occur due to the spread of middle ear infection to the inner ear, generalized convulsive seizures are unlikely and should be attributed to a brain event. This section requires further consideration. Even if the surgery is performed without issues, high-pressure cleaning in the presence of a skull base bone defect may provoke such occurrences. In human surgery, pressure irrigation is avoided in the middle ear and mastoid cavities, and only saline is used for sufficient cleaning without cytotoxic liquids.

**Do you want your identity to be public for this peer review?** For information about this choice, including consent withdrawal, please see our Privacy Policy

Reviewer #1: No

---

## [Author Response · Author response to Decision Letter 1]

25 Dec 2025

For reviewer #1

Suggestion: As discussed, the operational difficulty with the freehand frameless navigation system is influenced by the target size and its distance from the skin, which should be presented. From the preoperative CT scans, the following details should be included:

Response: Thank you for your kind suggestions.

Suggestion: List the maximum horizontal diameter of the target bulla, and maximum straight diameter relative to it on horizontal CT, and the maximum diameter of bulla on vertical CT if possible.

Response: According to this suggestion, in the revised version, we create Table 2, which includes CT measurements of the tympanic bulla’s size and the distance between the skin surface and the tympanic bulla. Additionally, in the Discussion section, the descriptions for the results are added.

Suggestion: If available, describe the distance from the skin surface to the inferior wall of the bulla.

Response: The distance between the skin surface and the tympanic bulla is measured in CT, and the result is written in Table 2.

Suggestion: Note any diagnoses of middle ear disease, such as acute phase of chronic otitis media or cholesteatoma otitis media infection, made through visual examination before or during surgery.

Response: The tympanic cavity could not be observed macroscopically through the skin opening during surgery. Almost all cases involved respiratory distress (pneumonia) and subsequently otitis media. These cases were considered to have chronic otitis media when treated with VBO at an average of 33.9 days after initial medical treatment. Thickening and extension in the tympanic bulla’s walls are the typical CT characteristics to show the chronic phase. These descriptions are added in lines 472-476 in the revised version.

Suggestion: The presence or absence of skull bone destruction and the extent of inflammation progression to surrounding tissues significantly impact the postoperative course. Therefore, these factors should be considered in the preoperative CT scan.

Response: Bone proliferation was assessed by measuring the increased width of the tympanic bulla’s wall on CT images (Table 2). Unfortunately, bone destruction (bone resorption) could not be evaluated because bone CT values (Hounsfield unit) were not measured in the cases examined. The CT examination could not assess the progression of inflammation into surrounding tissues.

Suggestion: On page 43, line 641: It is important to note that while dizziness may occur due to the spread of middle ear infection to the inner ear, generalized convulsive seizures are unlikely and should be attributed to a brain event. This section requires further consideration.

Response: We completely agree with this suggestion. Seizures are caused by mechanical damage to the inner ear structure and deeply located brain structure from a trocar used in the present VBO procedure. The sentences are changed and modified in lines 694-702 in the revised version.

Suggestion: Even if the surgery is performed without issues, high-pressure cleaning in the presence of a skull base bone defect may provoke such occurrences. In human surgery, pressure irrigation is avoided in the middle ear and mastoid cavities, and only saline is used for sufficient cleaning without cytotoxic liquids.

Response: In irrigation within the tympanic bulla, ozonated water was poured by manually pushing the piston of a 20 ml syringe via a catheter introduced through a trocar. The manual procedure to pour ozonated water was carefully performed. Thus, we believe that the pressure was not high within the cavity of the tympanic bulla during intra-ear irrigation. Because the pathogen was unknown at the time of medical treatments, ozonated water has been used in the present VBO procedure. Ozonated water inactivates immediately upon contact with biological tissues, leaving no residual irritants. However, we agree with the opinion that “only saline is used for sufficient cleaning without cytotoxic liquids”. Thus, in the revised version, the sentences are corrected in lines 615-637, by adding one new reference paper.

---

## [Decision Letter · Decision Letter 1]

26 Jan 2026

Utility of a freehand frameless navigation system in computed tomography-assisted ventral bulla osteotomy for otitis media in calves

PONE-D-25-45935R1

Dear Dr. Tsuka,

We’re pleased to inform you that your manuscript has been judged scientifically suitable for publication and will be formally accepted for publication once it meets all outstanding technical requirements.

Kind regards,

Toru Miwa

Academic Editor

PLOS One

Additional Editor Comments (optional):

Reviewers' comments:

Reviewer's Responses to Questions

**Comments to the Author**

Reviewer #1: (No Response)

2. Is the manuscript technically sound, and do the data support the conclusions?

Reviewer #1: Yes

3. Has the statistical analysis been performed appropriately and rigorously?

Reviewer #1: Yes

4. Have the authors made all data underlying the findings in their manuscript fully available?

Reviewer #1: Yes

5. Is the manuscript presented in an intelligible fashion and written in standard English?

Reviewer #1: Yes

Reviewer #1: (No Response)

**Do you want your identity to be public for this peer review?** For information about this choice, including consent withdrawal, please see our Privacy Policy

Reviewer #1: No

---

## [Editor Report · Acceptance letter]

PONE-D-25-45935R1

PLOS One

Dear Dr. Tsuka,

I'm pleased to inform you that your manuscript has been deemed suitable for publication in PLOS One. Congratulations! Your manuscript is now being handed over to our production team.

Kind regards,

on behalf of

Dr. Toru Miwa

Academic Editor

PLOS One